# RECURRENT NORMALIZATION PROPAGATION

**César Laurent, Nicolas Ballas & Pascal Vincent**[*]
Montreal Institute for Learning Algorithms (MILA)
Département d'Informatique et de Recherche Opérationnelle
Université de Montréal
Montréal, Québec, Canada
`{firstname.lastname}@umontreal.ca`

## ABSTRACT

We propose an LSTM parametrization that preserves the means and variances of the hidden states and memory cells across time. While having training benefits similar to Recurrent Batch Normalization and Layer Normalization, it does not need to estimate statistics at each time step, therefore, requiring fewer computations overall. We also investigate the parametrization impact on the gradient flows and present a way of initializing the weights accordingly.

We evaluate our proposal on language modelling and image generative modelling tasks. We empirically show that it performs similarly or better than other recurrent normalization approaches, while being faster to execute.

## 1 INTRODUCTION

Recurrent neural network have shown remarkably good performances for sequential modelling tasks including machine translation (Bahdanau et al., 2015), visual captioning (Xu et al., 2015; Yao et al., 2015) or question answering (Hermann et al., 2015). However, such models remain notoriously hard to train with gradient backpropagation. As the number of time steps in the input sequence increases, the contractive or expanding effects associated with the state-to-state transformation at each time step can shrink or grow exponentially, leading respectively to vanishing or exploding gradients (Hochreiter, 1991; Bengio et al., 1994; Pascanu et al., 2012). In particular, with gradient vanishing, states at a given time are not influenced by changes happening much earlier in the sequence, preventing the model from learning long-term dependencies.

While the long-term dependencies problem is unsolvable in absolute (Hochreiter, 1991; Bengio et al., 1994), different RNN parameterizations, such as LSTM or GRU (Hochreiter & Schmidhuber, 1997; Cho et al., 2014) can help mitigate it. Furthermore, the LSTM parametrization has been recently extended to include layer-wise normalization (Cooijmans et al., 2016; Ba et al., 2016), building upon Batch Normalization (BN) (Ioffe & Szegedy, 2015). By normalizing the hidden state distributions to a fix scale and shift through the different time steps, normalized LSTMs have been shown to ease training, resulting in a parametrization that converges faster than a standard LSTM.

However, normalized LSTM introduces extra-computations as it involves standardizing the hidden states, enforcing their means and variances at each time step. By contrast, we propose an LSTM reparametrization that allows by construction to cheaply preserve the normalization of the hidden states through time. Our approach can be seen as the recurrent counterpart to the recent normalization propagation applied in feed-forward network (Arpit et al., 2016). It results in faster training convergence similar to Layer Normalization (LN) and Recurrent Batch Normalization while requiring fewer operations per time step and generalizing naturally to variable length sequences.

In addition, we investigate the impact of our parametrization, and more generally of normalized LSTM, on the vanishing and exploding gradient problems. We observe that layer-wise normalization provides a direct way to orient LSTM behaviour toward either gradient explosion or vanishing, and therefore biases the LSTM either towards reliably storing bits of information throughout time or allowing it to be more sensitive to new input changes.

---

[*]Associate Fellow, Canadian Institute For Advanced Research (CIFAR)

We empirically validate our proposal on character-level language modelling on the Penn Treebank corpus (Marcus et al., 1993) and on image generative modelling, applying our normalisation to the DRAW architecture (Gregor et al., 2015).

The paper is structured as follows: section 2 provides a brief overview of the Batch-Normalized LSTM, in section 3 we derive our Normalized LSTM, section 4 investigates the impact of such normalization on the gradient flow, section 5 presents some experimental results, and we conclude in section 5.

## 2 PRE-REQUISITES

### 2.1 BN-LSTM

Batch-Normalized Long Short-Term Memory (BN-LSTM) (Cooijmans et al., 2016) is a reparametrization of LSTM that takes advantage of Batch Normalization (BN) to address the *Co-variate Shift* (Shimodaira, 2000) occurring between time steps. Changes in the LSTM output at one time-step are likely to cause correlated changes in the summed inputs of the sequence next time-steps. This *Temporal Covariate Shift* can slow down the training process as the parameters of the model must not only be updated to minimize the cost of the task at hand but also adapt to the changing distribution of the inputs. In other words, the latter time steps in an LSTM need to account for the shifting distribution of the previous hidden states.

BN-LSTM proposes to reduce this temporal covariate shift by fixing the mean and the variance at each time step, relying on the BN transform

$$\text{BN}(\mathbf{x}; \gamma, \beta) = \gamma \odot \frac{\mathbf{x} - \widehat{\mathbb{E}}[\mathbf{x}]}{\sqrt{\widehat{\text{Var}}[\mathbf{x}] + \epsilon}} + \beta \tag{1}$$

where $\widehat{\mathbb{E}}[\mathbf{x}], \widehat{\text{Var}}[\mathbf{x}]$ are the activation mean and variance estimated from the mini-batch samples. Given an input sequence $\mathbf{X} = (\mathbf{x}_1, \mathbf{x}_2, \ldots, \mathbf{x}_T)$, the BN-LSTM defines a sequence of hidden states $\mathbf{h}_t$ and memory cell states $\mathbf{c}_t$ according to

$$\begin{pmatrix} \tilde{\mathbf{i}}_t \\ \tilde{\mathbf{f}}_t \\ \tilde{\mathbf{o}}_t \\ \tilde{\mathbf{g}}_t \end{pmatrix} = \text{BN}(\mathbf{W}_x \mathbf{x}_t; \gamma_x, \beta_x) + \text{BN}(\mathbf{W}_h \mathbf{h}_{t-1}; \gamma_h, \beta_h) + \mathbf{b} \tag{2}$$

$$\mathbf{c}_t = \sigma(\tilde{\mathbf{i}}_t) \odot \tanh(\tilde{\mathbf{g}}_t) + \sigma(\tilde{\mathbf{f}}_t) \odot \mathbf{c}_{t-1} \tag{3}$$

$$\mathbf{h}_t = \sigma(\tilde{\mathbf{o}}_t) \odot \tanh(\text{BN}(\mathbf{c}_t; \gamma_c, \beta_c)), \tag{4}$$

where $\mathbf{W}_h \in \mathbb{R}^{d_h \times 4d_h}, \mathbf{W}_x \in \mathbb{R}^{d_x \times 4d_h}, \mathbf{b} \in \mathbb{R}^{4d_h}$ and the initial states $\mathbf{h}_0 \in \mathbb{R}^{d_h}, \mathbf{c}_0 \in \mathbb{R}^{d_h}$ are model parameters. $\sigma$ is the logistic sigmoid function, and $\odot$ denotes the Hadamard product. Ba et al. (2016) latter extended this parametrization by estimating the normalizing statistics $(\widehat{\mathbb{E}}[\mathbf{x}], \widehat{\text{Var}}[\mathbf{x}])$ using the different feature channels rather than mini-batch samples in order to naturally generalize to variable length sequences.

### 2.2 NORMALIZATION PROPAGATION

While increasing the training convergence speed relatively to a standard LSTM (Cooijmans et al., 2016), BN-LSTM needs to perform more computations per sample as it requires to compute 3x the BN transform at each time step.

On the other hand, Normalization Propagation (Norm Prop) (Arpit et al., 2016) aims at preserve the normalization of the input throughout the network. Unlike BN, the normalization doesn't rely on the statistics of the mini-batch. Instead, it is the structure of the network itself that maintains the normalization. We therefore propose an LSTM reparametrization that preserves the normalization through the different time steps in order to avoid those extra computation.

## 3    NORMALIZED LSTM

While Norm Prop properties are appealing for recurrent models, its application to LSTM is not straightforward due to the memory cell structure. In this section we show how to derive a LSTM reparametrization that preserves normalization of the state $\mathbf{h}_t$ through time.

### 3.1    CONSTRUCTION OF THE NORMALIZED LSTM

Following (Arpit et al., 2016; Salimans & Kingma, 2016), we will attempt to ensure, through an analytical reparametrization, that several intermediate quantities in the computation remain approximately standardized. We first compensate for the distribution changes induced by the weight matrices in the gates and cell candidate $\mathbf{g}_t$ computations

$$
\begin{pmatrix} \tilde{\mathbf{i}}_t \\ \tilde{\mathbf{f}}_t \\ \tilde{\mathbf{o}}_t \\ \tilde{\mathbf{g}}_t \end{pmatrix} = \gamma_x \frac{\mathbf{W}_x}{||\mathbf{W}_{x,i}||_2}\mathbf{x}_t + \gamma_h \frac{\mathbf{W}_h}{||\mathbf{W}_{h,i}||_2}\mathbf{h}_{t-1} + \mathbf{b}.
$$  (5)

where $||\mathbf{W}_{\cdot,i}||_2$ is the vector of L2-norm of each line of the matrix and $\gamma_x$ and $\gamma_h$ are the trainable rescaling factors that restore the representation power lost in the rescaling of the weight matrices. To preserve the constant error carousel mechanism of the LSTM, we use the usual cell update,

$$
\mathbf{c}_t = \sigma(\tilde{\mathbf{i}}_t) \odot \tanh(\tilde{\mathbf{g}}_t) + \sigma(\tilde{\mathbf{f}}_t) \odot \mathbf{c}_{t-1}
$$  (6)

Let us now construct an approximate analytical estimate of $\mathrm{Var}(\mathbf{c}_t)$. The evolution of $\mathbf{c}_t$ through time can bee seen as a geometric series, with $\sigma(\tilde{\mathbf{f}}_t)$ as constant ratio. Since $\sigma(\cdot)$ is upper-bounded by (and in practice smaller than) 1, $\mathbf{c}_t$ will converge in expectation to a fixed value. This is the reason why in BN-LSTM the mini-batch statistics converge to a fixed value after a few time steps (Cooijmans et al., 2016). Moreover, if we consider that $\tilde{\mathbf{i}}_t, \tilde{\mathbf{f}}_t, \tilde{\mathbf{g}}_t$ and $\mathbf{c}_{t-1}$ are (as a rough approximation) independent[1], we can use the variance product rule of two independent random variables $X$ and $Y$

$$
\mathrm{Var}[XY] = \mathrm{Var}[X]\mathrm{Var}[Y] + \mathrm{Var}[X]\mathbb{E}[Y]^2 + \mathrm{Var}[Y]\mathbb{E}[X]^2
$$  (7)

to compute $\mathrm{Var}[\mathbf{c}_t]$. Considering that $\mathbb{E}[\tanh(\tilde{\mathbf{g}}_t)] \approx 0$ and assuming that the cell has converged *i.e.* $\mathrm{Var}[\mathbf{c}_t] = \mathrm{Var}[\mathbf{c}_{t-1}]$, we have

$$
\mathrm{Var}[\mathbf{c}_t] = \mathrm{Var}[\tanh(\tilde{\mathbf{g}}_t)]\frac{\mathrm{Var}[\sigma(\tilde{\mathbf{i}}_t)] + \mathbb{E}[\sigma(\tilde{\mathbf{i}}_t)]^2}{1 - \mathrm{Var}[\sigma(\tilde{\mathbf{f}}_t)] - \mathbb{E}[\sigma(\tilde{\mathbf{f}}_t)]^2}
$$  (8)

We can therefore analytically or numerically compute the mean and variance of each of those elements, assuming that both input $\mathbf{x}_t$ and hidden state $\mathbf{h}_{t-1}$ are independent drawn from $\mathcal{N}(0,1)$

$$
\mathbb{E}[\mathbf{i}_t] = \mathbb{E}[\sigma(\gamma_x z_x + \gamma_h z_h)]
$$  (9)

$$
\mathrm{Var}[\mathbf{i}_t] = \mathrm{Var}[\sigma(\gamma_x z_x + \gamma_h z_h)]
$$  (10)

$$
\mathbb{E}[\mathbf{g}_t] = \mathbb{E}[\tanh(\gamma_x z_x + \gamma_h z_h)]
$$  (11)

$$
\mathrm{Var}[\mathbf{g}_t] = \mathrm{Var}[\tanh(\gamma_x z_x + \gamma_h z_h)]
$$  (12)

where $z_x, z_h \sim \mathcal{N}(0,1)$. The statistics of the gates $\mathbf{o}_t$ and $\mathbf{f}_t$ can be computed in a similar way. We can then compute the value to which $\mathrm{Var}[\mathbf{c}_t]$ converges. Using this variance estimate, we compensate $\mathbf{c}_t$ in order to compute the next hidden state $\mathbf{h}_t$

$$
\mathbf{h}_t = \sigma(\tilde{\mathbf{o}}_t) \odot \tanh\left(\frac{\gamma_c \mathbf{c}_t}{\sqrt{\mathrm{Var}[\mathbf{c}_t]}}\right)
$$  (13)

Since we assumed that $\mathrm{Var}[\mathbf{h}_{t-1}] = 1$, to ensure that we need to correct for the variance induced by the product of the $\tanh$ with the output gate. Using again the variance product rule (equation 7) we obtain

$$
\mathrm{Var}[\mathbf{h}_t] = \mathrm{Var}\left[\tanh\left(\frac{\gamma_c \mathbf{c}_t}{\sqrt{\mathrm{Var}[\mathbf{c}_t]}}\right)\right](\mathrm{Var}[\sigma(\tilde{\mathbf{o}}_t)] + \mathbb{E}[\sigma(\tilde{\mathbf{o}}_t)]^2)
$$  (14)

We can estimate this variance through similar computation than equation 12. Scaling $\mathbf{h}_t$ with $1/\sqrt{\mathrm{Var}[\mathbf{h}_t]}$ ensure that its variance is 1 and so the propagation is maintained throughout the recurrence.

---

[1]This assumption is strong, but we don't have any easy way to model the covariance between those terms without estimating it from the data.

## 3.2 PROPOSED REPARAMETRIZATION

Using equations 5, 6 and 13, we propose the following reparametrization of the LSTM, simply called the *Normalized LSTM*

$$
\begin{pmatrix} \tilde{\mathbf{i}}_t \\ \tilde{\mathbf{f}}_t \\ \tilde{\mathbf{o}}_t \\ \tilde{\mathbf{g}}_t \end{pmatrix} = \gamma_x \frac{\mathbf{W}_x}{||\mathbf{W}_{x,i}||_2} \mathbf{x}_t + \gamma_h \frac{\mathbf{W}_h}{||\mathbf{W}_{h,i}||_2} \mathbf{h}_{t-1} + \mathbf{b}
\tag{15}
$$

$$
\mathbf{c}_t = \sigma(\tilde{\mathbf{i}}_t) \odot \tanh(\tilde{\mathbf{g}}_t) + \sigma(\tilde{\mathbf{f}}_t) \odot \mathbf{c}_{t-1}
\tag{16}
$$

$$
\mathbf{h}_t = \frac{1}{\sqrt{\mathrm{Var}[\mathbf{h}_t]}} \left[ \sigma(\tilde{\mathbf{o}}_t) \odot \tanh \left( \frac{\gamma_c \mathbf{c}_t}{\sqrt{\mathrm{Var}[\mathbf{c}_t]}} \right) \right]
\tag{17}
$$

where $\mathrm{Var}[\mathbf{c}_t]$ and $\mathrm{Var}[\mathbf{h}_t]$ are computed using equations 8 and 14, respectively. Those two variances are estimated at the initialization of the network (eq. 10 to eq. 12), and are then kept fixed during the training as in Norp Prop. $\gamma_x$, $\gamma_h$ and $\gamma_c$ are parameters learned via gradient descent.

Note that the reparametrization of equation 15 is identical to Weight Normalization (Weight Norm) (Salimans & Kingma, 2016). The main difference comes from equation 17, where we compensate for the variance of $\mathbf{c}_t$, the $\tanh$ and $\sigma(\tilde{\mathbf{o}}_t)$, which ensures a normalized propagation. Overall, this reparametrization is equivalent in spirit to the BN-LSTM, but it benefits from the same advantages that Norm Prop has over BN: There is no dependence on the mini-batch size and the computation is the same for training and inference. Also, the rescaling of the matrices $\mathbf{W}_x$ and $\mathbf{W}_h$ can be done before the recurrence, leading to computation time closer to a vanilla LSTM.

## 3.3 WEIGHTS INITIALIZATION

With such reparametrization of the weight matrices, one can think that the scale of the initialization of the weights doesn't matter in the learning process anymore. It is actually true for the forward and backward computation of the layer

$$
y_i = \frac{a\mathbf{W}_i}{||a\mathbf{W}_i||_2} x = \frac{\mathbf{W}_i}{||\mathbf{W}_i||_2} x
\tag{18}
$$

$$
\frac{\partial y_i}{\partial x} = \frac{a\mathbf{W}_i}{||a\mathbf{W}_i||_2} = \frac{\mathbf{W}_i}{||\mathbf{W}_i||_2}
\tag{19}
$$

and since the variance of both forward and backward passes is fixed, using an initialization scheme such as Glorot (Glorot & Bengio, 2010) doesn't make sense with Norm Prop. However, the update of the parameters is affected by their scale:

$$
\frac{\partial y_i}{\partial \mathbf{W}_{ij}} \frac{\partial \mathcal{L}}{\partial y_i} = \frac{1}{||\mathbf{W}_i||_2} \left[ x_j - y_i \frac{\mathbf{W}_{ij}}{||\mathbf{W}_i||_2} \right] \frac{\partial \mathcal{L}}{\partial y_i}
\tag{20}
$$

The scale of the parameters affect the learning rate of the layer: the bigger the weights, the smaller the update. This induces a regularization effect in Norm Prop that is also present in BN (Ioffe & Szegedy, 2015). However, this could possibly be an issue for such parametrization: different initializations lead to different learning rates, and it is true even with adaptive step rules, such as Adam (Kingma & Ba, 2014). Moreover, the parameters that are not normalized (such as $\gamma$ and $\mathbf{b}$) aren't affected by this effect, and so they are not regularized. This is the reason why forcing the weight matrices to have a unit L2 norm of the lines, as proposed in Arpit et al. (2016), helps the training procedure.

To still benefit from the reduction of the learning rate, which is know to ease the optimization (Vogl et al., 1988), we propose to simply force the unit L2 norm of the lines of the matrices and combine it with a global learning rate decay schedule.

## 4 GRADIENT PROPAGATION IN NORMALIZED LSTM

In this section we study the gradient flow in the Normalized LSTM. Since this reparametrization is similar to the BN-LSTM, the analysis we do here can be transposed to the BN-LSTM case.

### 4.1 The Exploding and Vanishing Gradients Problem

Given an input sequence $\mathbf{X} = (\mathbf{x}_1, \mathbf{x}_2, \ldots, \mathbf{x}_T)$, we consider a recurrent network, parametrized by $\theta$, that defines a sequence of hidden states $\mathbf{h}_t = f_\theta(\mathbf{h}_{t-1}, \mathbf{x}_t)$ and cost function $\mathcal{L}$ which evaluates the model performance on a given task. Such network is usually trained using backpropagation through time, where the backpropagation is applied on the time-unrolled model. The chain rule can be applied in order to compute the derivative of the loss $\mathcal{L}$ with respect to parameters $\theta$.

$$\frac{\partial \mathcal{L}}{\partial \theta} = \sum_{1 \le t \le T} \frac{\partial \mathcal{L}_t}{\partial \theta} = \sum_{1 \le t \le T} \sum_{1 \le k \le t} \frac{\partial \mathcal{L}_t}{\partial \mathbf{h}_k} \frac{\partial \mathbf{h}_k}{\partial \mathbf{h}_t} \frac{\partial \mathbf{h}_t}{\partial \theta}. \tag{21}$$

The factors $\frac{\partial \mathbf{h}_k}{\partial \mathbf{h}_t} = \prod_{k \le l \le t} \frac{\partial \mathbf{h}_l}{\partial \mathbf{h}_{l-1}}$ transports the error "in time" from step $t$ back to step $k$ and are also the cause of vanishing or exploding gradient in RNN (Pascanu et al., 2012). Indeed, if the Jacobian $\frac{\partial \mathbf{h}_l}{\partial \mathbf{h}_{l-1}}$ has singular value different from 1, the factor $\frac{\partial \mathbf{h}_k}{\partial \mathbf{h}_t}$, which is a product of $t - k$ Jacobian matrices will either explode or vanish.

### 4.2 Gradient of the Normalized LSTM

To study the gradient propagation of the Normalized LSTM, we first need to derive it. Using equation 15-17, we can write the gradient of $\mathbf{h}_t$ with respect to $\mathbf{h}_{t-1}$

$$\mathbf{a}_t = \frac{1}{\sqrt{\mathrm{Var}[\mathbf{h}_t]}} \tanh\left(\frac{\gamma_c \mathbf{c}_t}{\sqrt{\mathrm{Var}[\mathbf{c}_t]}}\right) \tag{22}$$

$$\frac{\partial \mathbf{h}_t}{\partial \mathbf{h}_{t-1}} = \frac{\partial \mathbf{o}_t}{\partial \mathbf{h}_{t-1}} \odot \mathbf{a}_t + \mathbf{o}_t \odot \frac{\partial \mathbf{a}_t}{\partial \mathbf{h}_{t-1}} \odot \left[\frac{\partial \mathbf{i}_t}{\partial \mathbf{h}_{t-1}} \odot \mathbf{g}_t + \mathbf{i}_t \odot \frac{\partial \mathbf{g}_t}{\partial \mathbf{h}_{t-1}} + \frac{\partial \mathbf{f}_t}{\partial \mathbf{h}_{t-1}} \odot \mathbf{c}_{t-1}\right] \tag{23}$$

As we can see in equation 23 with the normalization, the gradient depends not only on the derivative of the cell candidate, the gates and the output *tanh*, but also on on the variance of $\mathbf{h}_t$ and $\mathbf{c}_t$.

If we assume that $\mathbf{h}_{t-1}$ and $\mathbf{x}_t$ are independent, we can compute the variance of $\mathbf{c}_t$. Neglecting the weight matrices and the effect of the gates, we can write from equations 8 and 14

$$\mathrm{Var}[\mathbf{c}_t] \approx \mathrm{Var}[\mathbf{g}_t] = \mathrm{Var}[\tanh(z)], \quad z \sim \mathcal{N}(0, \gamma_x^2 + \gamma_h^2) \tag{24}$$

$$\mathrm{Var}[\mathbf{h}_t] = \mathrm{Var}[\tanh(z)], \quad z \sim \mathcal{N}(0, \gamma_c^2(\gamma_x^2 + \gamma_h^2)) \tag{25}$$

In both cases, the variance depends explicitly on the value of the different $\gamma$: The bigger the $\gamma$, the higher the variance. Neglecting again the weight matrices, we can now write the equations of the cell candidates $\mathbf{g}_t$ and the gates $\mathbf{i}_t, \mathbf{o}_t$ and $\mathbf{f}_t$ with respect to $\mathbf{h}_{t-1}$

$$\frac{\partial \mathbf{g}_t}{\partial \mathbf{h}_{t-1}} = \frac{\partial \tanh(\tilde{\mathbf{g}}_t)}{\partial \tilde{\mathbf{g}}_t} \frac{\partial \tilde{\mathbf{g}}_t}{\partial \mathbf{h}_{t-1}} = \left(1 - \tanh(\gamma_x \mathbf{x}_t + \gamma_h \mathbf{h}_{t-1})^2\right) \gamma_h \tag{26}$$

$$\frac{\partial \mathbf{i}_t}{\partial \mathbf{h}_{t-1}} = \frac{\partial \sigma(\tilde{\mathbf{i}}_t)}{\partial \tilde{\mathbf{i}}_t} \frac{\partial \tilde{\mathbf{i}}_t}{\partial \mathbf{h}_{t-1}} = \sigma(\gamma_x \mathbf{x}_t + \gamma_h \mathbf{h}_{t-1})(1 - \sigma(\gamma_x \mathbf{x}_t + \gamma_h \mathbf{h}_{t-1}))\gamma_h \tag{27}$$

The gradients of $\mathbf{o}_t$ and $\mathbf{f}_t$ can be computed similarly. The effect of the $\gamma$ here is double: They appear both in the activation function, where they control the saturation regime, and $\gamma_h$ also appears as a multiplicative term in the gradient. They should therefore be small enough to prevent the activation from saturating too much, but at the same time $\gamma_h$ can't be too small, because it can also make the gradients vanish. Putting it all together, we have

$$\frac{\partial \mathbf{h}_t}{\partial \mathbf{h}_{t-1}} = \frac{\partial \mathbf{o}_t}{\partial \tilde{\mathbf{o}}_t} \gamma_h \odot \mathbf{a}_t + \mathbf{o}_t \odot \frac{\partial \mathbf{a}_t}{\partial \tilde{\mathbf{a}}_t} \frac{\gamma_c}{\sqrt{\mathrm{Var}[\mathbf{c}_t]}} \odot \gamma_h \left[\frac{\partial \mathbf{i}_t}{\partial \tilde{\mathbf{i}}_t} \odot \mathbf{g}_t + \mathbf{i}_t \odot \frac{\partial \mathbf{g}_t}{\partial \tilde{\mathbf{g}}_t} + \frac{\partial \mathbf{f}_t}{\partial \tilde{\mathbf{f}}_t} \odot \mathbf{c}_{t-1}\right] \tag{28}$$

In this equations we can see that the different $\gamma$ directly scale the gradient, and they also control the saturation of the activation functions. Bad initialization of $\gamma$ could thus lead to saturation or explosion regimes. Figure 1 shows the norm of the gradient with respect to $\gamma_x$ and $\gamma_h$ in a simulated LSTM. As we can see, one important parameter is the ratio between $\gamma_h$ and $\gamma_x$: They control most of the propagation of the gradients. If $\gamma_x > \gamma_h$, the network will focus more on the input and so the gradients will tend to vanish more. On the other hand, if $\gamma_h > \gamma_x$, the network will tend have less vanishing gradients, but will focus less on its inputs.

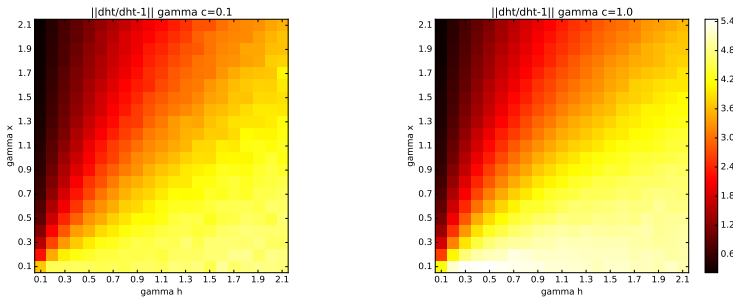

Figure 1: Norm of the gradients for one time step in an LSTM with respect to $\gamma_x$ and $\gamma_h$ (simulation). Left: $\gamma_c = 0.1$. Right: $\gamma_c = 1.0$.

## 5   EXPERIMENTS

### 5.1   CHARACTER-LEVEL LANGUAGE MODELLING

The first task we explore is character-level language modelling on the Penn Treebank corpus (Marcus et al., 1993). The goal is to predict the next character of the sequence given the previous ones. We use the same splits as Mikolov et al. (2012) and the same training procedure as Cooijmans et al. (2016), i.e. we train on sequences of length 100, with random starting point. The model is a 1000 units LSTM followed by a Softmax classifier. We use orthogonal initialization for the weight matrices. Because Norm Prop requires normalized inputs, we multiply the one-hot inputs vector with an untrained but fixed orthogonal matrix. This tricks does not only help the optimization of Norm Prop, but also all other variants.

To compare the convergence properties of Norm Prop against LN and BN, we first ran experiments using Adam (Kingma & Ba, 2014) with learning rate 2e-3, exponential decay of 1e-3 and gradient clipping at 1.0. As explained in section 3.3, we rescale the matrices such that they have a unit norm on the lines. For Norm Prop, we use $\gamma_x = \gamma_h = 2$ and $\gamma_c = 1$, for LN all the $\gamma = 1.0$ and for BN all the $\gamma = 0.1$. The results are presented in Table 1 and in Figure 2.

| Model | Validation | Time |
|---|---|---|
| Baseline | 1.455 | **386** |
| Weight Norm | 1.438 | 402 |
| Batch Norm | 1.433 | 545 |
| Layer Norm | 1.439 | 530 |
| Norm Prop | **1.422** | 413 |

Table 1: Perplexity (bits-per-character) on sequences of length 100 from the Penn Treebank validation set, and training time (seconds) per epoch.

To show the potential of Norm Prop against other state-of-the-art system, we followed Ha et al. (2016) and apply dropout on both the input and output layer ($p = 0.1$) and recurrent dropout inside the LSTM ($p = 0.1$). We also used the *Batch Data Normalization* scheme presented by Arpit et al. (2016), so we standardize each input example using the mini-batch statistics and use population statistics at inference time. Finally, we also reduce the learning rate decay to 1e-4, to compensate for the fact that a network with dropout needs more time to train. The results are presented in Table 2.

As we can see in Figure 2 and in Table 1, Norm Prop compares really well against the other reparametrization. Also Norm Prop is roughly 30 % computationally faster[2] than BN and LN. LN shows better optimization performances, but also overfits more. We also see that both optimization and generalization are better than the ones from Weight Norm, which shows the importance of compensating for the variance of $c_t$ and $h_t$. Moreover, although Norm Prop doesn't combine well with

---

[2]The GPU used is a NVIDIA GTX 750.

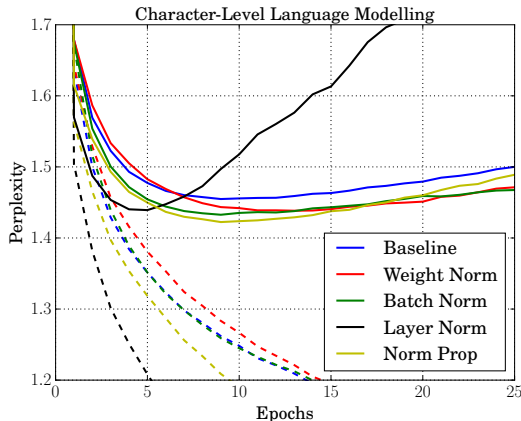

Figure 2: Perplexity (bits-per-character) on sequences of length 100 from the Penn Treebank corpus. The dashed lines are the training curves, and the solid ones are the validation curves.

| Model | Test |
|---|---|
| Recurrent Dropout LSTM (Semeniuta et al., 2016) | 1.301 |
| Zoneout LSTM (Krueger et al., 2016) | 1.27 |
| Layer Norm LSTM (Ha et al., 2016) | 1.267 |
| HyperLSTM (Ha et al., 2016) | 1.265 |
| Norm Prop LSTM (ours) | 1.262 |
| Layer Norm HyperLSTM (Ha et al., 2016) | **1.250** |

Table 2: Perplexity (bits-per-character) of the full Penn Treebank test sequence.

dropout in feed-forward networks (Arpit et al., 2016), it works will with recurrent dropout, as we can see in Table 2. We believe it is because recurrent dropout is less affecting its output distribution than dropout in feed forward networks, because we copy the variable at the previous time step instead of setting it to 0. With such regularization, Norm Prop compares well with other state-of-the-art approaches.

## 5.2 DRAW

The second task we explore is a generative modelling task on binarized MNIST (Larochelle & Murray, 2011) using the *Deep Recurrent Attentive Writer* (DRAW) (Gregor et al., 2015) architecture. DRAW is a variational auto-encoder, where both encoder and decoder are LSTMs, and has two attention mechanisms to select where to read and where to write.

We use Jörg Bornschein's implementation[3], with the same hyper-parameters as Gregor et al. (2015), ie the read and write size are 2x2 and 5x5 respectively, the number of glimpses is 64, the LSTMs have 256 units and the dimension of $z$ is 100. We use Adam with learning rate of 1e-2, exponential decay of 1e-3 and mini-batch size of 128. We use orthogonal initialization and force the norm of the lines of the matrices to be 1. For Norm Prop, we use $\gamma_x = \gamma_h = \gamma_c = 0.5$. The test variational bound for the first 100 epochs is presented in Figure 3.

As we can see in Figure 3, both Weight Norm and Norm Prop outperform the baseline network by a significant margin. Also, as expected, Norm Prop performs better than Weight Norm, showing one again the importance of the compensation of the variance of $\mathbf{c}_t$ and $\mathbf{h}_t$. Table 3 shows the test variational bound after 200 epochs of training. Norm Prop also compares favorably against LN.

---

[3] https://github.com/jbornschein/draw

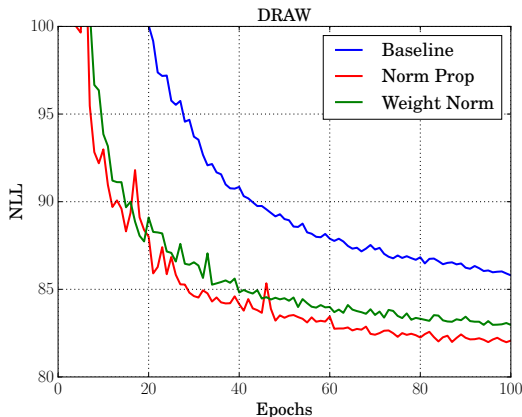

Figure 3: Test negative log-likelihood on binarized MNIST.

| Model | DRAW |
|---|---|
| Baseline (ours) | 84.30 |
| Layer Norm (Ba et al., 2016) | 82.09 |
| Weight Norm (ours) | 81.98 |
| Norm Prop (ours) | **81.17** |

Table 3: Test variational log likelihood (nats) after 200 epochs of training.

## 6 CONCLUSION

Based on the BN-LSTM, we have shown how to build a Normalized LSTM that is able to preserve the variance of its output at each time step, by compensating for the variance of the cell and the hidden state. Such LSTM can be seen as the Norm Prop version of the BN-LSTM, and thus benefits from the same advantages that Norm Prop has over BN, while being way faster to compute. Also, we propose a scheme to initialize the weight matrices that takes into account the reparametrization. Moreover, we have derived the gradients of this LSTM and pointed out the importance of the initialization of the rescaling parameters. We have validated the performances of the Normalized LSTM on two different tasks, showing similar performances than BN-LSTM and LN-LSTM, while being significantly faster in computation time. Also, unlike the feed-forward case, this architecture works well with recurrent dropout, leading to close to state-of-the-art performances on the character-level language modelling task.

Future work includes trying this architecture on more challenging tasks and also studying the impact of not keeping the variance estimates of the cell and the hidden states fixed during the learning process.

### ACKNOWLEDGMENTS

Part of this work was funded by Samsung. We used Theano (Theano Development Team, 2016), Blocks and Fuel (van Merriënboer et al., 2015) for our experiments. We also want to thanks Caglar Gulcehre and Tim Cooijmans for the talks and Jörg Bornschein for his DRAW implementation.

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
