# Peer review of "Recurrent Normalization Propagation"

_ICLR 2017 — rejected_

[Official Review · AnonReviewer3 · rating 6 · confidence 3 · 16 Dec 2016]
**No Title**

The paper proposes an extension of weight normalization / normalization propagation to recurrent neural networks. Simple experiments suggest it works well.

The contribution is potentially useful to a lot of people, as LSTMs are one of the basic building blocks in our field.

The contribution is not extremely novel: the change with respect to weight normalization is minor. The experiments are also not very convincing: Layer normalization is reported to have higher test error as it overfits on their example, but in terms of optimization it seems to work better. Also the authors don't seem to use the data dependent parameter init for weight normalization as proposed in that paper.

[Official Review · AnonReviewer2 · rating 6 · confidence 4 · 18 Dec 2016]
**incremental**

I think this build upon previous works, in the attempt of doing something similar to batch norm specific for RNNs. To me the experiments are not yet very convincing, I think is not clear this works better than e.g. Layer Norm or not significantly so. I'm not convinced on how significant the speed up is either, I can appreciate is faster, but it doesn't feel like order of magnitude faster. The theoretical analysis also doesn't provide any new insights. 

All in all I think is good incremental work, but maybe is not yet significant enough for ICLR.

[Official Review · AnonReviewer1 · rating 4 · confidence 4 · 20 Dec 2016]
**Sloppy writing, unsufficient experimental validation**

The authors show how the hidden states of an LSTM can be normalised in order to preserve means and variances. The method’s gradient behaviour is analysed. Experimental results seem to indicate that the method compares well with similar approaches.

Points

1) The writing is sloppy in parts. See at the end of the review for a non-exhaustive list.

2) The experimental results show marginal improvements, of which the the statistical significance is impossible to asses. (Not completely the author’s fault for PTB, as they partially rely on results published by others.) Weight normalisation seems to be a viable alternative in the: the performance and runtime are similar. The implementation complexity of weight norm is, however, arguably much lower. More effort could have been put in by the authors to clear that up. In the current state, practitioners as well as researchers will have to put in more effort to judge whether the proposed method is really worth it for them to replicate.

3) Section 4 is nice, and I applaud the authors for doing such an analysis.


List of typos etc.

- maintain -> maintain
- requisits -> requisites
- a LSTM -> an LSTM
- "The gradients of ot and ft are equivalent to equation 25.” Gradients cannot be equivalent to an equation.
- “beacause"-> because
- One of the γx > γh at the end of page 5 is wrong.

[Final Decision · Program Chairs · 06 Feb 2017]
**ICLR committee final decision**

Paper proposes a modification of batch normalization. After the revisions the paper is a much better read. However it still needs more diverse experiments to show the success of the method.
 
 Pros:
 - interesting idea with interesting analysis of the gradient norms
 - claims to need less computation
 
 Cons:
 - Experiments are not very convincing and only focus on only a small set of lm tasks.
 - The argument for computation gain is not convincing and no real experimental evidence is presented. The case is made that in speech domain, with long sequences this should help, but it is not supported.
 
 With more experimental evidence the paper should be a nice contribution.